# Structure and Glass Transition Temperature of Amorphous Dispersions of Model Pharmaceuticals with Nucleobases from Molecular Dynamics

**DOI:** 10.3390/pharmaceutics13081253

**Published:** 2021-08-13

**Authors:** Ctirad Červinka, Michal Fulem

**Affiliations:** Department of Physical Chemistry, University of Chemistry and Technology, Prague, Technická 5, CZ-166 28 Prague, Czech Republic; fulemm@vscht.cz

**Keywords:** active pharmaceutical ingredients, amorphous dispersion, glass transition, molecular dynamics

## Abstract

Glass transition temperature (*T*_g_) is an important material property, which predetermines the kinetic stability of amorphous solids. In the context of active pharmaceutical ingredients (API), there is motivation to maximize their *T*_g_ by forming amorphous mixtures with other chemicals, labeled excipients. Molecular dynamics simulations are a natural computational tool to investigate the relationships between structure, dynamics, and cohesion of amorphous materials with an all-atom resolution. This work presents a computational study, addressing primarily the predictions of the glass transition temperatures of four selected API (carbamazepine, racemic ibuprofen, indomethacin, and naproxen) with two nucleobases (adenine and cytosine). Since the classical non-polarizable simulations fail to reach the quantitative accuracy of the predicted *T*_g_, analyses of internal dynamics, hydrogen bonding, and cohesive forces in bulk phases of pure API and their mixtures with the nucleobases are performed to interpret the predicted trends. This manuscript reveals the method for a systematic search of beneficial pairs of API and excipients (with maximum *T*_g_ when mixed). Monitoring of transport and cohesive properties of API–excipients systems via molecular simulation will enable the design of such API formulations more efficiently in the future.

## 1. Introduction

Low solubility of crystalline active pharmaceutical ingredients (API) in water, often being hydrophobic molecules, represents a challenge to be faced in the development of most new drugs. To overcome this issue, API are often present in their amorphous solid phases in drug formulations [1,2]. As such amorphous phases are thermodynamically metastable and tend to crystallize, which diminishes the solubility again, artificial stabilization of amorphous API is often required to benefit from their higher solubility over a relevant time horizon [3]. Solid mixtures (dispersions) of API and polymers have been subject to extensive research recently [4,5]. Polymers are known to form amorphous solid phases rather than crystals, suggesting that this dispersion strategy should yield a long-term stability of amorphous API. On the other hand, synthetic polymers can have a negative impact on the treated patients. Therefore, co-amorphous mixtures of the API with suitable low-molecular-weight natural compounds (sugars, amino acids, nucleobases, etc.), acting as excipients [6], are becoming the primary subject of research in this field [7,8]. Such a strategy is similar to using cryoprotective biomolecules for stabilization of biochemical samples at extreme conditions [9].

Fundamental characteristics of amorphous solids are the structure relaxation time [10] and the temperature of their glass transition, *T*_g_, being a second-order phase transition between a rubbery (liquid) phase and a glassy (solid) phase. To assess the stability of an amorphous solid at the given conditions (temperature, pressure, humidity), one has to consider the difference in the storing temperature or the temperature in vivo. The closer we get to the *T*_g_, the higher the expected tendency of the solid to crystallize. A model problem is thus to maximize the *T*_g_ for the given API by finding suitable excipient(s) and optimizing the composition of such mixtures.

Empirical relationships are often used to estimate *T*_g_, usually exploiting the fusion temperature (*T*_f_), stating that *T*_g_ usually ranges from 0.6 *T*_f_ to 0.8 *T*_f_ for most pharmaceutical chemicals [11]. There are also simplifying rules for estimation of the *T*_g_ for mixtures, e.g., Gordon–Taylor [12] and Fox [13] equations, employing the densities of present species apart from their *T*_g_. However, a priori knowledge of such data can be a limiting obstacle. Clearly, molecular structure, its symmetry, packing ability, and interactions with the other components are the factors governing the tendency of glass formation, or contrarily, crystallinity. Predominantly, large molecular weight, massive conformational flexibility, uniform distribution of electronegative atoms, and hydrogen bonding are the factors favoring the existence of the glassy state, while molecular symmetry or aromaticity usually promote crystallinity [14]. Calorimetric and spectroscopic studies [15] as well as molecular-dynamics investigations [16] suggest that distinct trends of the vitrification tendency among similar chemicals can be traced to the strength of intermolecular interactions in bulk materials.

Molecular dynamics (MD) simulations represent a viable pathway with a solid physical background for predictions of *T*_g_ for any chemical systems at arbitrary conditions [17,18]. Liquid and amorphous solid phases differ in the mobility of their molecules by definition. Such a dynamical property can be directly studied by MD, evaluating the molecular mean-squared displacement (MSD) over the simulated trajectory. Investigating the temperature dependence of the MSD enables us to clearly discriminate between the two dynamical regimes (diffusive liquid and sub-diffusive glassy). MSD typically rises more steeply with temperature for the liquids and a clear break point of the MSD slope can be identified at *T*_g_ [19]. Unfortunately, to suppress the numerical noise and to obtain a sufficiently smooth temperature dependence of the MSD, sufficiently large molecular ensembles have to be simulated over sufficiently long trajectories, causing the higher cost of such MD predictions of *T*_g_. Therefore, cheaper alternatives are often exploited within MD, focusing on volumetric data only, which are accessible reliably from considerably shorter trajectories than MSD [20]. Either the direct dependence of the system volume on temperature *V*(*T*) or the response rate of the system volume to sudden temperature drops can be used for predictions of *T*_g_. Again, liquids usually exhibit a steeper volume increase with temperature. Such a volumetric MD approach typically works well for mixtures with polymers. Nevertheless, localization of the break point of the simulated *V*(*T*) data set can be ambiguous for low-molecular-weight mixtures, for which analyzing MSD and volumetric data simultaneously seems important.

In this study, structural and transport properties and vitrification of four model API (carbamazepine, RS-ibuprofen, indomethacin, and naproxen) and two nucleobases (adenine and cytosine) and their mixtures of various compositions are investigated using molecular dynamics. Presented simultaneous analysis of transport and cohesive properties of bulk phases of API–excipient dispersions is an illustration of how an efficient design of API formulations can work in the future. All our simulations for the pure compounds serve for benchmarking the computational accuracy of our model, whereas the predicted properties of the mixtures with no available experimental data are intended to narrow the scope of our future investigations of this topic.

## 2. Computational Methods

### 2.1. Force-Field Models

Molecular structure and interactions of all bulk phases were modeled using classical non-polarizable all atom force-field OPLS models, the parameters of which were mostly culled from previous literature studies [21,22,23]. Sources of individual force-field parameter sets are given in Table 1. A summary of non-bonded force-field parameters, used throughout this work, is given in the Appendix A. Missing bonding parameters and corresponding force constants for API molecules were taken from original OPLS parametrization for model organic species [24], taking the chemical similarities of the present atomic types into account. The literature force field for naproxen [22] uses a united-atom model not treating the hydrogen atoms explicitly, which led us to reparametrization of its atomic charges. For the nucleobases, we adopted the original OPLS parametrization by the Jorgensen group [25], updating the atomic charges via the CHELPG procedure. Unlike for the API, the MP2/aug-cc-pVTZ level of theory was used in the case of nucleobases, since the B3LYP function artificially exaggerates the planarity of amino groups attached to the cores of nucleobase molecules. All quantum–chemical calculations of the force-field parameters were performed in Gaussian 16 [26].

Tautomer equilibrium in liquid nucleobases can be hardly verified experimentally. According to the ab initio electronic energies of adenine [27] and cytosine [28] tautomers in the vapor phase, the presence of most tautomers can be ruled out due to their unfavorably high energies. In the case of cytosine, the tautomer in its crystal and the most stable vapor-phase tautomer do not match. Bulk phases of both nucleobases were constrained to contain a single tautomer that is present in the given crystalline phase, since these tautomers can be expected to form beneficial hydrogen bonds also in the liquid. Relevant tautomers of all studied molecules are depicted in Figure 1.

### 2.2. Molecular Dynamics

Molecular dynamics simulations were performed in the LAMMPS code [29] (version 31 October 2015). Simulation boxes were created from around 600–700 molecules, counting over 15,000 atoms in total, placing the molecules to random positions in a cubic box by the Packmol code [30]. Apart from pure compounds, 1:3 (molar), 1:1, and 3:1 mixtures of API and nucleobases were also simulated. Such simulation boxes were then simulated as an *NPT* ensemble using the velocity Verlet integrating scheme [31], a Nosé–Hoover thermostat, and a barostat [32], maintaining the temperature and pressure at 450 K and 100 KPa, respectively. Particle–particle particle–mesh (PPPM) long-range electrostatics solver [33] and the integral dispersion tail corrections were applied. The SHAKE method [34] was used to constrain the lengths of all chemical bonds terminating in hydrogen atoms. MD simulations under the given setup were performed with a time step of 1 fs and the simulated properties were sampled each 1 ps. An initial equilibration period at 450 K spanned 1 ns and was followed by additional 1 ns equilibrations at the individual temperatures in the range 200–500 K with a step of 10 K. Production runs spanned a 5 ns period at each temperature, which represents a compromise balancing the computational complexity and trajectory length for dissecting reasonable self-diffusivity data.

## 3. Results and Discussion

### 3.1. Force-Field Validation

The availability of experimental crystal structures of pure nucleobases is very limited, comprising usually single-temperature X-ray diffraction results [35,36]. The availability of crystal structures for model API is somewhat better [37,38,39,40,41]. To benchmark the accuracy of the underlying force fields for predictions of phase behavior, we primarily selected polymorphs, which are the most stable at ambient conditions. To increase the validity of our force-field benchmarking, we included also the α form of indomethacin, which is probably metastable at the given conditions [42]. There are also unique experimental data on liquid-phase densities for racemic ibuprofen, indomethacin, and naproxen [43], which all proved stable above their melting temperatures. Altogether, the existing experimental data on densities allow us to gain some insight into the consistency and accuracy of structural parameters of all used force-field models. The results of this initial structural validation, consisting of MD simulations of densities of the bulk phases of simulated compounds (liquid and crystals for API and crystals for nucleobases), are given in Table 2. 

Simulated densities of the crystals are in a perfect agreement with experimental data, with the root-mean squared error (RMSE) amounting to 1.7%. On the other hand, a looser agreement of the liquid-phase densities was observed with RMSE at 5.5%. Such a trend could be explained by the increased complexity of quantitative capturing of the bulk-phase structure, which is more strongly affected by thermal expansion at elevated temperatures. Still, such an accuracy in terms of simulated bulk-phase densities can be accepted as satisfying for our analyses of the radial distribution functions (e.g., in Appendix A).

A more stringent test for the force-field quality can be reached through benchmarking of enthalpic properties. Due to the very low volatility of all studied API and nucleobases, experimental data on vaporization or sublimation enthalpies that could be used for this purpose are extremely rare, as such materials are extremely rare. While no such data exist for the nucleobases, carbamzepin, and indomethacin to our knowledge, we can illustrate the computational accuracy for naproxen and racemic ibuprofen. Based on the sublimation pressure measurements, calorimetric enthalpy of fusion and qualified estimates of the difference in heat capacities of gas and condensed phases, experimental vaporization enthalpy of naproxen at 410 K can be evaluated as 106.9 ± 3.0 kJ mol^−1^ [44]. Assuming the ideal behavior of the vapor, our simulations predict the corresponding value at 108.2 ± 1.0 kJ mol^−1^, which can be accepted as a very close agreement of theory with experiment, with the actual difference of 1.3 kJ mol^−1^ well within the chemical accuracy threshold (~4 kJ mol^−1^). Similarly, vaporization enthalpy for racemic ibuprofen at 410 K amounts to 84.4 ± 3.0 kJ mol^−1^, based on vapor pressure measurements [45], the calorimetric heat capacity of liquid [42], and the heat capacity of ideal gas computed in this work using the rigid rotor–harmonic oscillator model with molecular paramters and double-scaled frequencies [46] obtained at the B3LYP/6-31G(d) level of theory in Gaussian [26]. In this case, our simulation predicts a value 90.0 kJ mol^−1^, not fitting within the chemical accuracy, but being reasonably close to the experiment.

Another option for benchmarking the energetic force-field parameters is represented by the enthalpies of fusion, which are available at least for all the studied API [42]. Melting temperatures of the nucleobases are expected to range above the upper thermal stability threshold of such molecules [42]. Table 3 lists a comparison of simulated and experimental fusion enthalpies. Simulated enthalpies of fusion exhibit an RMSE of 32% (corresponding to 11 kJ∙mol^−1^) with preferred underestimation of fusion enthalpies by MD. Such a result is far from any quantitative accuracy, but corresponds well to the typical accuracy of MD simulations of the fusion enthalpy organic compounds [42]. 

### 3.2. Vitrification Analysis

Simulated densities and self-diffusivities were plotted as functions of temperature to apply the trend shift method of determination of the glass transition temperature. Graphical illustrations of this data processing are given in the Appendix A. In general, simulated densities exhibit only a minor scatter and their temperature trends are concave and relatively smooth, possibly rendering the localization of the trend shift between the expansion regimes of the glass and liquid phases somewhat ambiguous. Contrarily, self-diffusivities usually exhibit a clear trend shift at temperatures high enough for the bulk phase to become fluid. On the other hand, the nature of the transport properties and the relatively short trajectories (5 ns) that were simulated at each temperature cause a higher scatter of the temperature trends of the self-diffusivities. Furthermore, reaching a linear regime (in terms of molecular mean-squared displacements) for the glass phase becomes rather coincidental within a short molecular-dynamics trajectory, especially for the lowest temperatures.

Taking these aspects into account, we combined the trend-shift analysis of simulated density and self-diffusivity data. Final predicted glass transition temperatures were then taken as the average of these two approaches. Note that the density-derived *T*_g_ and the self-diffusivity-derived *T*_g_ for a given system did not differ by more than 25 K (even less for one-component systems), which complies with the standard deviations of *T*_g_ (amounting to 24 K) estimated through the error propagation law from the standard deviations of the interpolation parameters, describing the temperature dependencies of density and self-diffusivity of the simulated systems. 

To our knowledge, there are no experimental data on *T*_g_ for pure nucleobases and for most of the mixtures relevant for this predictive study. Table 4 compares the simulated *T*_g_ with experimental data, which reveals a relatively large systematic overestimation of *T*_g_ within the current computational model. Mean deviation of the calculated *T*_g_ amounts to 67 K, which is comparable in magnitude with the reported *T*_g_ value for indomethacin [23], was obtained by a gradual cooling of the melt within the simulation. On the other hand, such a deviation of simulated *T*_g_ is substantially larger than that observed for comparable non-polarizable simulations of ionic liquids (around 20 K) [47]. Similarly low accuracy of predited *T*_g_ can be observed also for the equimolar mixture of adenine with indomethacin. Its experimental *T*_g_ amounts to 327.3 K [48], whereas our model predicts the value 366.7 K. 

While the underlying force fields, which we used in our previous work for ionic liquids [47] and in this work for API, are similarly sophisticated (both all-atom non-polarizable OPLS) and yield comparably accurate structural properties and phase change enthalpies [49,50], the different character of cohesive forces binding the molecules in the bulk phase can possibly explain such a discrepancy in the accuracy of *T*_g_. Ionic liquids are bound by strong, yet directionless electrostatic interactions and their massive cohesion impedes diffusion of their ions (excessively in MD). Contrarily, the cohesion of bulk phases of API is governed by site-specific anisotropic hydrogen bonding, the anisotropic influence of which on the transport properties might be even more complex to capture within the simulations. As a result, we observe a larger error of simulated *T*_g_ for API.

Figure 2 displays the predicted *T*_g_ for all simulated API, nucleobases, and their mixtures. Keeping in mind the limited quantitative accuracy of the simulated *T*_g_ values, only qualitative conclusions can be drawn based on these data. Our MD simulations captured the ranking of the API in terms of their *T*_g_ values. Ibuprofen is correctly predicted to exhibit the lowest *T*_g_, followed by naproxen, while carbamazepine and indomethacin exhibit the largest and very close *T*_g_. Both considered nucleobases in their pure bulk phase are predicted to exhibit higher *T*_g_ than any included API. Such a relationship naturally leads to a gradual increase in *T*_g_ of API–nucleobase mixtures as the molar fraction of the nucleobases rises. This trend is most pronounced for ibuprofen and naproxen as their *T*_g_ are the lowest in our test set. Any scatter of the *T*_g_ versus composition data points in Figure 2 should be primarily attributed to the computational noise rather than to any imprint of irregular behavior.

Next, we analyzed the trends of the self-diffusivities of API molecules in the individual simulated systems. We selected a universal temperature of 410 K for this analysis as all the considered API appear to be liquid in our simulations at such conditions. Note that all these materials were reported to be chemically stable at 410 K, as evidenced by the observed melting temperatures above 410 K [42], or by the density measurements of liquid API performed up to 450 K [43]. Figure 3 reveals that the compound with the lowest *T*_g_, ibuprofen, also exhibits the highest self-diffusivities in our simulations, being at least two orders of magnitude larger than for the remaining compounds. Increase in the *T*_g_ of mixtures of ibuprofen with the nucleobases is accompanied by a steep decline in self-diffusivities of ibuprofen molecules in such mixtures. Ibuprofen molecules are an order of magnitude less mobile in the 3:1 adenine–ibuprofen mixture than in pure liquid ibuprofen at 410 K. Such a decline is even more pronounced when ibuprofen is mixed with cytosine. A similar, yet less significant behavior can be seen also for carbamazepine and naproxen. Except for indomethacin, the presence of the nucleobase acts in favor of stabilization of the amorphous API, limiting the diffusive degrees of freedom of its molecules. On the other hand, both nucleobases are predicted to act as weak fluidifiers for indomethacin, slightly increasing its self-diffusivities in such mixtures, which remain the lowest among all simulated systems anyway.

Figure 4 shows trends of the simulated molar volumes of the API–nucleobase mixtures. Most systems exhibit behavior very close to an ideal mixture with minimum excess volumes. The most significant excess volumes are observed for 1:3 ibuprofen–nucleobase mixtures, amounting to −1.0% and −1.5% of the total molar volume of the mixture in the presence of adenine and cytosine, respectively. Such a negative excess volume indicates the possibility of such mixtures to improve their steric packing efficiency, which arises from the ibuprofen–nucleobase interactions. Although this phenomenon is compatible with the steepest diffusivity decline observed for the mixtures of ibuprofen, such a percentage of molar volumes is comparable in magnitude with the computational uncertainty that is typical for the simulated molar volumes.

The presence of a carboxyl group in the given API molecules (or an amide group in carbamazepine) opens the possibilities for massive hydrogen bonding in the bulk phases of API. Figure 5 compares the radial distribution functions representing the closest contacts of the most significant contributors to the hydrogen bonding in pure API. The closest dimer of the carboxyl groups, held together by an antisymmetric pair of hydrogen bonds, can be identified for all three API containing the COOH group. Among these, indomethacin forms the strongest carboxyl dimer in the liquid, evidenced by the closest contact O–H···O distance of only 1.50 Å and a massive amplitude of the first peak of the respective radial distribution function. These closest dimers in pure ibuprofen and naproxen are somewhat looser with the O–H···O contact distance of 1.75 Å. Carbamazepine, missing a complete COOH group, lacks the possibility of forming the closest carboxyl dimer, resulting in its closest O–H···O contact at 2.25 Å. 

Interestingly, the presence of the nucleobases, which also interact through strong hydrogen bonds, can severely disrupt the closest API–carboxyl dimers. This can be best observed in Figure 5 for the case of ibuprofen. Its O–H···O radial distribution function alters its shape upon mixing with the nucleobases, significantly amplifying the latter peaks that are due to more remote H···O contacts, corresponding to rather looser chains of carboxyl groups. Additionally, saturation of the most beneficial acceptor site of API molecules, their carbonyl oxygen atom, with a hydrogen bond from a nucleobase forces the pairs of API to interact also with their hydroxyl groups, which essentially imprints also in the former O–H···O radial distribution function, only at larger separations. Such a phenomenon cannot be observed in the mixtures of carbamazepine as there are no closest dimers to be disrupted.

Individual studied API significantly differ in their capabilities to act as hydrogen bond donors or acceptors when interacting with the nucleobases. Figure 6 displays the most important interaction types, for which the API molecule is the hydrogen acceptor. For all four API, the carbonyl oxygen atoms present in the carboxyl group are the primary acceptor sites. Based on the contact distances and peak amplitudes in Figure 6, the ability of accepting a hydrogen bond wanes in the order: Cbz > Ind > Nap > Ibu. Adenine is then a somewhat more efficient hydrogen bond donor than cytosine. In the case of adenine, its N–H moiety from the five-membered ring is the prevailing bond donor, while for cytosine, both the ring-incorporated N–H and the amino NH_2_ groups contribute practically equivalently to hydrogen bonding to API. These aspects are illustrated in the Appendix A.

When the ability of the four API to donate a hydrogen bond is compared, a decreasing tendency can be observed in the order: Ind > Ibu > Nap > Cbz, see Figure 7. Clearly, the API carboxyl hydrogen atoms are the most actively donated ones, in stark contrast to practically no hydrogen bond donation from the amino group of carbamazepine. Cytosine molecules accept the hydrogen bonds from API significantly more efficiently than adenine due to the carbonyl oxygen atom attached to the cytosine ring. For adenine, the nitrogen atom in the six-membered ring in the opposite position to the amino group is the most important hydrogen bond acceptor.

Mixing carbamazepine with the nucleobases leads to the formation of stronger hydrogen bonding to that occurring in the pure bulk carbamazepine, as the O_Z_···H_A/C1_ contact distances are 0.5 Å shorter than the original O_Z_···H_Z2_ ones. Mixed with the remaining three API, only cytosine seems to be capable of matching the importance of the original O_API_···H_API_ interactions. This can be traced to the O–C–N–H moiety of cytosine, enabling the formation of strong clusters with the carboxyl groups of the API, held again by a pair of hydrogen bonds. This structural feature probably does not reach such an importance in the case of adenine, since its N–C–N–H moieties form weaker hydrogen bonds than cytosine does.

Cohesive interactions in a bulk liquid can be monitored by the vaporization energy or enthalpy. Figure 8 reveals that the vaporization energies, and thus the magnitudes of the cohesion, are mostly consistent with the trends of the predicted self-diffusivities. The most diffuse liquid, ibuprofen, exhibits also the lowest vaporization energy and vice versa for indomethacin, which is a significant outlier in terms of its vaporization energy. Cohesion in bulk ibuprofen is appreciably weaker than in both considered nucleobases, leading to an increase in the vaporization energies of its mixtures, supporting its stabilization in the amorphous state. The remaining two nucleobases, carbamazepine and naproxen, exhibit similar vaporization energies as both nucleobases.

The outlying position of indomethacin can be also caused by its largest molecular size among the considered molecules. When the trends of vaporization energies are expressed in terms of specific energies (related per mass unit), both nucleobases significantly outperform all the API in terms of their cohesion (0.72 J∙g^−1^ for adenine and 0.98 J∙g^−1^ for cytosine vs. 0.43, 0.42, 0.53, and 0.46 J∙g^−1^ for carbamazepine, ibuprofen, indomethacin, and naproxen, respectively). Still, bulk indomethacin is the most strongly bound API also in terms of the specific vaporization energies.

## 4. Conclusions

We performed molecular dynamics simulations of four selected active pharmaceutical ingredients and their mixtures with low-molecular-weight biocompatible excipients. Although the present simulations offer a fair accuracy of structural or enthalpic data of bulk amorphous systems, indicating that the underlying force field are well-balanced, such models are far from reaching a quantitative accuracy in terms of the predicted glass transition temperatures of individual systems. Still, valuable insight on the structure, interactions, and dynamics of important chemical compounds and pharmaceutically relevant systems was gained. Trends of the predicted glass transition temperatures among individual species were captured correctly, which enabled a qualitative interpretation of the relationships between the vitrification, diffusivity, and strength of hydrogen bonding. Chemical reactivity among individual active pharmaceutical ingredients and excipients, which is expected to strongly affect the vitrification, could not be principally captured in the current classical simulations. This work focused on the contribution of non-covalent interactions to vitrification, and it presents a method to design efficient excipients for particular active ingredients in order to maximize the glass transition temperature. To improve the predictions of the glass transition temperatures of complex pharmaceuticals, more sophisticated force fields, e.g., including atomic polarizability, need to be developed, or affordable quantum–chemical levels of theory (currently, at least the semi-empirical ones) have to substitute the classical force fields in molecular-dynamics simulations.

## Figures and Tables

**Figure 1 pharmaceutics-13-01253-f001:**
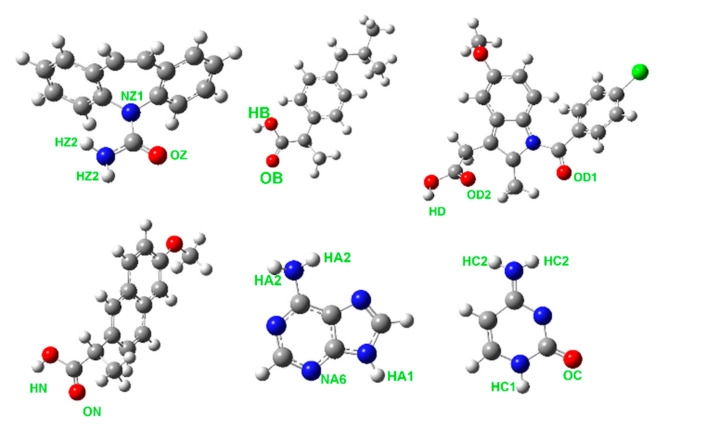
Molecular structures of carbamazepine (**top left**), ibuprofen (**top center**), and indomethacin, (**top right**) and naproxen (**bottom left**), adenine (**bottom center**), and cytosine (**bottom right**). Atom types contributing to the hydrogen bonding most strongly are marked for each molecule.

**Figure 2 pharmaceutics-13-01253-f002:**
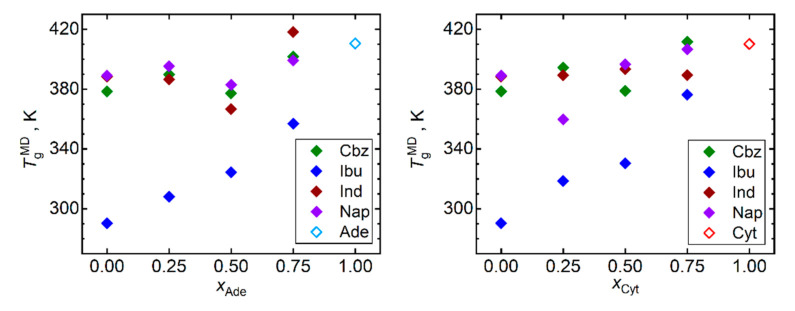
Glass transition temperatures *T*_g_ obtained by the trend shift method from MD simulations of densities and diffusivities (results averaged) of mixtures of nucleobases (adenine, Ade—left; and cytosine, Cyt—right) with API (carbamazepine, Cbz; RS-Ibuprofen, Ibu; Indomethacin, Ind; and naproxene, Nap).

**Figure 3 pharmaceutics-13-01253-f003:**
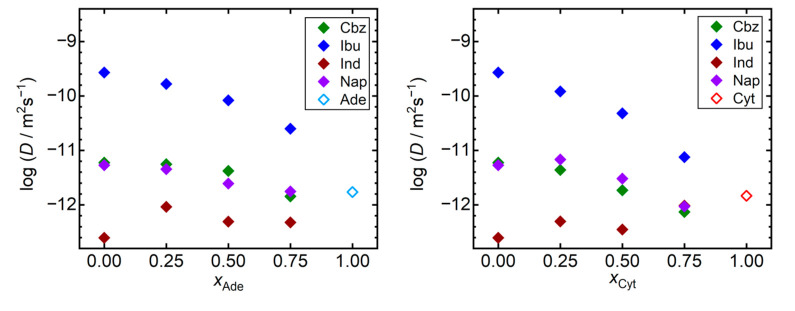
Self-diffusivities (*D*) of API molecules at 410 K (full symbols) and nucleobases (empty symbols) in the liquid. Mixtures of nucleobases (adenine, Ade—left; and cytosine, Cyt—right) with API (carbamazepine, Cbz; RS-Ibuprofen, Ibu; Indomethacin, Ind; and naproxene, Nap) are characterized by the molar fraction of the nucleobases (*x*_Ade_, *x*_Cyt_). Data are shown on the logarithmic scale for a better readability.

**Figure 4 pharmaceutics-13-01253-f004:**
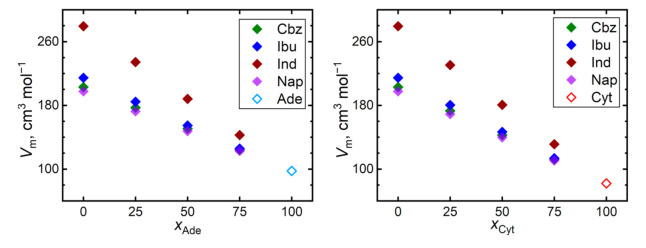
Molar volumes of mixtures of the nucleobases (adenine, Ade—left; and cytosine, Cyt—right) with API (carbamazepine, Cbz; RS-Ibuprofen, Ibu; Indomethacin, Ind; and naproxene, Nap) at 410 K.

**Figure 5 pharmaceutics-13-01253-f005:**
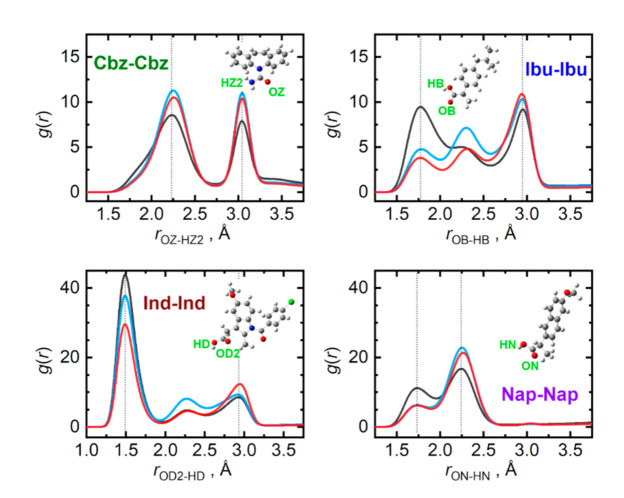
Hydrogen bonding among individual API molecules in the liquid phase at 410 K. Radial distribution functions are given for the most significant contacts in pure API (grey), and their equimolar mixtures with adenine (light blue) and cytosine (red).

**Figure 6 pharmaceutics-13-01253-f006:**
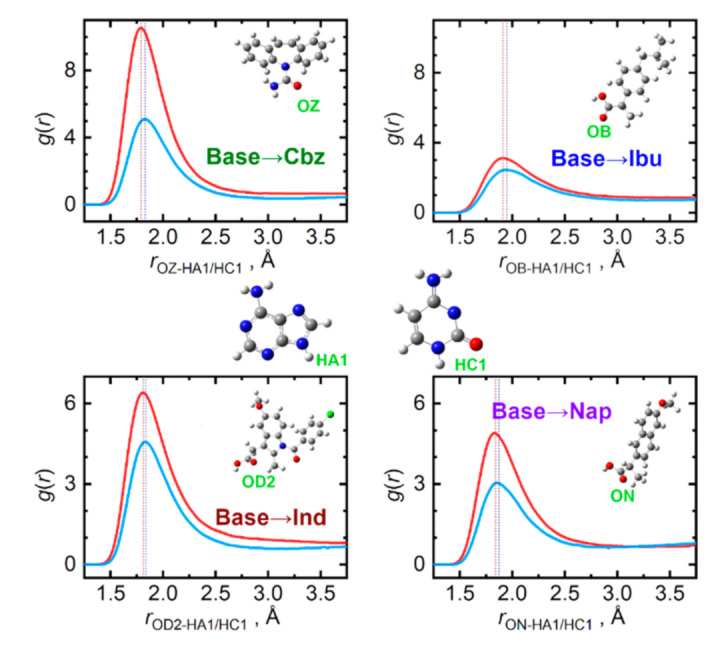
Hydrogen bonding between API (hydrogen acceptor) and nucleobase (hydrogen donor) molecules in the liquid phase at 410 K. Radial distribution functions are given for the most significant contacts in equimolar mixtures of API with adenine (light blue) and cytosine (red).

**Figure 7 pharmaceutics-13-01253-f007:**
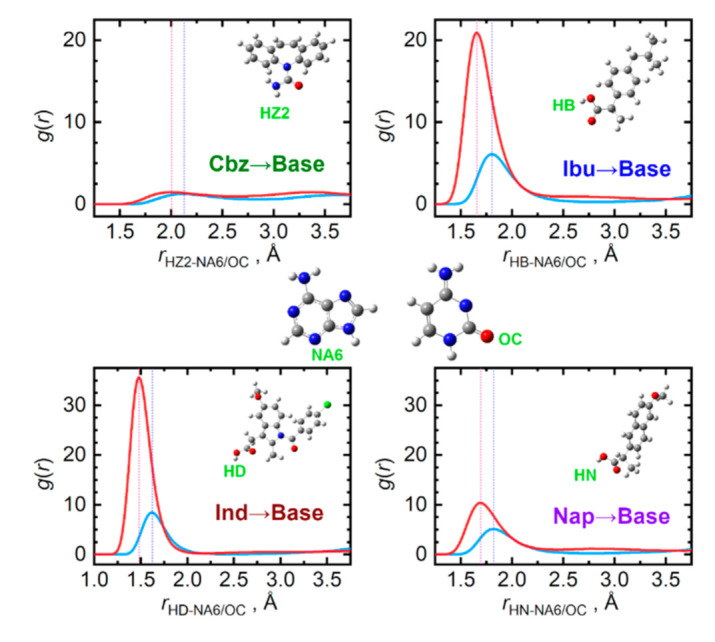
Hydrogen bonding between API (hydrogen donor) and nucleobase (hydrogen acceptor) molecules in the liquid phase at 410 K. Radial distribution functions are given for the most significant contacts in equimolar mixtures of API with adenine (light blue) and cytosine (red).

**Figure 8 pharmaceutics-13-01253-f008:**
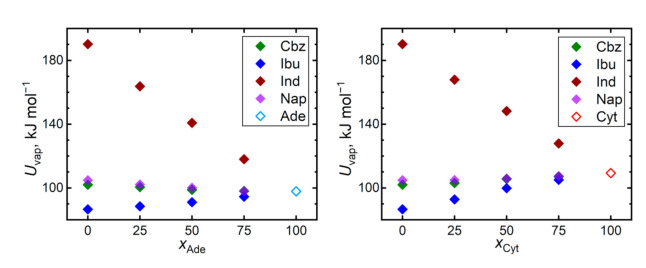
Vaporization energy *U*_vap_ required to evaporate 1 mole of the given liquids at 410 K.

**Table 1 pharmaceutics-13-01253-t001:** Overview of references to the original parametrizations of force fields for individual ions included in this work.

Term	Carbamazepine	Ibuprofen	Indomethacin	Naproxen	Adenine	Cytosine
Atomic charges	CHELPG, this work ^a^	Greiner et al. [21]	Xiang and Anderson [23]	CHELPG, this work ^a^	CHELPG, this work ^b^	CHELPG, this work ^b^
Dispersion	Original OPLS [24]	Greiner et al. [21]	Xiang and Anderson [23]	Römer and Kraska. [22]	Pranata et al. [25]	Pranata et al. [25]
Molecular geometry	QM, this work^a^	Greiner et al. [21]	Xiang and Anderson [23]	Römer and Kraska. [22]	Pranata et al. [25]	Pranata et al. [25]
Force constants	Original OPLS [24]	Greiner et al. [21]	Original OPLS [24]	Römer and Kraska. [22]	Pranata et al. [25]	Pranata et al. [25]

^a^ The underlying quantum level of theory was B3LYP/aug-cc-pVTZ; ^b^ The underlying quantum level of theory was MP2/aug-cc-pVTZ.

**Table 2 pharmaceutics-13-01253-t002:** Comparison of simulated and experimental bulk phase densities *ρ* (g∙cm^−3^).

Compound	Phase	Temperature, K ^a^	*ρ* _MD_	*ρ* _exp_	100(*ρ*_MD_/*ρ*_exp_^−1^)
Carbamazepine	Crystal III	293	1.335	1.333 [41]	0.1
Ibuprofen	Crystal I	296	1.115	1.117 [40]	−0.2
	Liquid	350	1.006	0.966 [43]	4.1
		400	0.968	0.924 [43]	4.7
Indomethacin	Crystal α	203	1.408	1.420 [39]	−0.9
	Crystal γ	120	1.418	1.401 [38]	1.2
	Liquid	400	1.284	1.231 [43]	4.3
		450	1.264	1.183 [43]	6.9
Naproxen	Crystal	293	1.308	1.263 [37]	3.6
	Liquid	430	1.154	1.088 [43]	6.1
		480	1.116	1.048 [43]	6.5
Adenine	Crystal	293	1.506	1.494 [35]	0.8
Cytosine	Crystal	293	1.537	1.502 [36]	−1.6

^a^ Experimental density determination was performed at this temperature.

**Table 3 pharmaceutics-13-01253-t003:** Comparison of simulated and experimental fusion enthalpies Δ_fus_*H* (kJ∙mol^−1^).

Compound	Phase	Temperature, K	Δ_fus_*H*_MD_	Δ_fus_*H*_exp_ ^a^	100(Δ_fus_*H*_MD_/Δ_fus_*H*_exp_^−1^)
Carbamazepin	Crystal III	293	30.6	27.4	12
Ibuprofen	Crystal I	296	15.1	26.4	−43
Indomethacin	Crystal γ	203	20.6	38.1	−46
Naproxen	Crystal	293	29.1	32.4	−10

^a^ Experimental data taken from ref. [42].

**Table 4 pharmaceutics-13-01253-t004:** Comparison of simulated (density-derived $T_{g,\mathrm{MD}}^{\rho }$, diffusivity-derived $T_{g,\mathrm{MD}}^{D}$, and averaged $T_{g,\mathrm{MD}}^{}$) and experimental (*T*_g,exp_) glass transition temperatures (given in K).

Compound	$$T_{g,\mathrm{MD}}^{\rho }$$	$$T_{g,\mathrm{MD}}^{D}$$	$$T_{g,\mathrm{MD}}^{}$$	*T* _g,exp_ ^a^	100(*T*_g,MD_/*T*_g,exp_ − 1)
Carbamazepin	384	374	379	315	20
Ibuprofen	295	286	290	228	27
Indomethacin	388	388	388	313	24
Naproxen	343	347	345	278	24

^a^ Experimental data taken from ref. [42].

## Data Availability

The data presented in this study are available in the Appendix A of this article.

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
