# Peer review of "Structure and Glass Transition Temperature of Amorphous Dispersions of Model Pharmaceuticals with Nucleobases from Molecular Dynamics"

_pharmaceutics, 2021, doi:10.3390/pharmaceutics13081253_

Round 1
Reviewer 1 Report
This paper describes application of molecular dynamics simulation for prediction of glass transition temperature and intermolecular interactions. This topic is worth of investigation and further development of these techniques with improvement in predictability should reduce number of unnecessary experiments in the formulation development. I consider that manuscript is well written, but there are several points that authors should consider in the revision.
- Some parts require language correction, particularly some uncommon terms should be replaced with more appropriate terms.
- It will be very useful if you can provide some experimental data regarding Tg of API-nucleobase pairs. It is reasonable to expect some discrepancies between experimental and predicted values, as MD predictions are still far from being accurate.
- The caption of Figure 3 is not clear and it is difficult to conclude what is actually presented on this figure. What do you consider under whole simulated system? Some symbols on this graph are not explained in legend or caption.
- I suggest authors to present chemical formula of interacting compounds on Figures 5, 6 and 7, like it is presented in Supporting material. This will be more convenient for readers.
- The sentence “This section may be divided by subheadings. It should provide a concise and precise description of the experimental results, their interpretation, as well as the experimental conclusions that can be drawn” should not be a part of the manuscript.
- Please reconsider the title of the manuscript.
Author Response
Rev #1:
This paper describes application of molecular dynamics simulation for prediction of glass transition temperature and intermolecular interactions. This topic is worth of investigation and further development of these techniques with improvement in predictability should reduce number of unnecessary experiments in the formulation development. I consider that manuscript is well written, but there are several points that authors should consider in the revision.
1. Some parts require language correction, particularly some uncommon terms should be replaced with more appropriate terms.
Ans: We thank for this comment. Although we are not sure what phrasing in particular the referee meant, we attempted to correct and clarify our phrasing throughout the manuscript.
2. It will be very useful if you can provide some experimental data regarding Tg of API-nucleobase pairs. It is reasonable to expect some discrepancies between experimental and predicted values, as MD predictions are still far from being accurate.
Ans: We agree that a larger set of experimental data would enable to improve the statistical significance of our comparison of the calculated and experimental Tg values. To assess the accuracy of the underlying MD simulations for Tg predictions, we selected the particular API molecules due to the availability of experimental Tg and other properties. Unfortunately, these data are extremely scarce for multi-component systems, including mixtures of API with biomolecules. Furthermore, our intention was to perform a computational study at first to identify promising candidate systems, thus to narrow the scope of subsequent experimental investigation of such amorphous dispersions, which we plan in next stages of our research of this topic. Fortunately, we already have a few unpublished experimental Tg data points which are relevant for this case. We added a comparison of our predictions with these new data and commented it in the manuscript accordingly.
3. The caption of Figure 3 is not clear and it is difficult to conclude what is actually presented on this figure. What do you consider under whole simulated system? Some symbols on this graph are not explained in legend or caption.
Ans: We thank for this remark enabling us to improve the readability of the Figure. The whole simulated system refers to the entire ensemble of API and nucleobases, with self-diffusivities averaged over both types of molecules. Finally, we decided to remove these averaged data from the plot, as these did not bring much new information, and possibly, made the plot less clear. We added explanation of all the symbols in the caption and reformulated it to be more comfortable for the reader.
4. I suggest authors to present chemical formula of interacting compounds on Figures 5, 6 and 7, like it is presented in Supporting material. This will be more convenient for readers.
Ans: Note that the legend of particular atom types was already depicted in Figure 1 of the main paper. Still, we decided to update the mentioned figures each with a graphic legend of the respective atomic sites in contact.
5. The sentence “This section may be divided by subheadings. It should provide a concise and precise description of the experimental results, their interpretation, as well as the experimental conclusions that can be drawn” should not be a part of the manuscript.
Ans: We thank the referee for this warning. Clearly, this was mistakenly not deleted upon typesetting into the journal template.
6. Please reconsider the title of the manuscript.
Ans: We updated the title to be more specific: Structure and Glass Transition Temperature of Amorphous Dispersions of Model Pharmaceuticals with Nucleobases from Molecular Dynamics.
Reviewer 2 Report
The submitted manuscript present the results of the computational studies on the binary mixtures of chosen APIs with two nucleotide basis. In the future, those results could be possibly used to predict the recrystallization of amorphous APIs in their mixtures with excipients and therefore to choose the most appropriate ones. The aim, which was the prediction of Tg, was not reached which was honestly stated in lines 352-353 by the Authors. While I really appreciate testing of the new methods in order to improve the design of the solid dosage forms, I think this work needs major revision. My comments are listed below.
Line 20, it should be „sophisticated”
The abstract is a little bit too long, please shorten it.
Line 31, the low solubility is not necessarily the effect of hydrophobicity. Usually they are connected, but not always.
Lines 33, 34, “medicaments” is repeated, please replace by i.e. drug
Line 60, how can “molecular weight” favor the existence of a glassy state? Do you mean “large molecular weight”?
Table 1, this is just a technical issue, you should try to fit “Dispersion” in one line
Line 102, please give more details, have you used Gaussian? Or other QM software?
Tables 2 and 3. I have noticed that you have chosen a particular polymorph for each API. Why those particular ones? Are they most stable? It should be explained in the manuscript.
Figures 2 and 3, the colors of Nap and Ibu are very similar, could you please change one of them for better clarity?
Line 263, it should be “ibuprofen”
Table 3, this is just a technical issue, you should try to fit “Temperature, K” in one line
Line 218, it should be “ibuprofen”
Figure 1, why only one enantiomer of ibuprofen is present here?
General question. I understand that you have wanted to test the new method of calculations in order to predict the Tg of binary systems (this was one of the aim of this work). So, why have you chosen the models for that the experimental data is limited? I.e., Figure 2. It would be more meaningful if you could compare those computational results with the corresponding experimental ones (measuring the Tg of the mixtures of those API with either Ade or Cys in different molar ratio). If you can’t compare this with the experimental results you don’t really know how far from the “true” values are you.
Line 206, it should be „sophisticated”
Line 208, the other possible explanation is that, to properly model the Tg, you need to perform the ab initio MD, which, for such large systems is currently computationally non affordable.
Lines 347-349, this should be deleted.
Figure 8, aren’t there any experimental data in the literature to compare those results?
Lines 364-365, or the application of some QM methods, at least semi-empirical.
Author Response
Rev #2:
The submitted manuscript present the results of the computational studies on the binary mixtures of chosen APIs with two nucleotide basis. In the future, those results could be possibly used to predict the recrystallization of amorphous APIs in their mixtures with excipients and therefore to choose the most appropriate ones. The aim, which was the prediction of Tg, was not reached which was honestly stated in lines 352-353 by the Authors. While I really appreciate testing of the new methods in order to improve the design of the solid dosage forms, I think this work needs major revision. My comments are listed below.
1. Line 20, it should be „sophisticated”.
Ans: Corrected.
2. The abstract is a little bit too long, please shorten it.
Ans: We thank for this opinion. We shortened the abstract by some 50 words.
3. Line 31, the low solubility is not necessarily the effect of hydrophobicity. Usually they are connected, but not always.
Ans: We rephrased the given introductory phrase, now admitting that not all API are hydrophobic, and not stating a direct link between the hydrophobicity and solubility.
4. Lines 33, 34, “medicaments” is repeated, please replace by i.e. drug
Ans: The word medicaments was replaced by drugs.
5. Line 60, how can “molecular weight” favor the existence of a glassy state? Do you mean “large molecular weight”?
Ans: Indeed, we intended to mention the positive correlation between the molecular weight and preferential glass formation. We corrected the sentence as the referee suggests.
6. Table 1, this is just a technical issue, you should try to fit “Dispersion” in one line.
Ans: We leave this final type-setting matter to the production office of the journal.
7. Line 102, please give more details, have you used Gaussian? Or other QM software?
Ans: Yes, all QM calculations were performed in Gaussian 16. We mentioned it in the paper.
8. Tables 2 and 3. I have noticed that you have chosen a particular polymorph for each API. Why those particular ones? Are they most stable? It should be explained in the manuscript.
Ans: Our initial criterion for polymorph selection was existing experimental data on the crystal structure and solid-liquid phase transitions. Naturally, both aspects are most often fulfilled concomitantly only for the most stable polymorphs at ambient conditions, which were our first choice for all our simulations. To increase the validity of our force-field benchmarking, we included also the α form of indomethacin, which is metastable at the given conditions. We mentioned this context in the manuscript.
9. Figures 2 and 3, the colors of Nap and Ibu are very similar, could you please change one of them for better clarity?
Ans: In our opinion, the color coding of blue (Ibu) and purple (Nap) is distinct enough. Note that other color shades (red, cyan) are used for the nucleobases, which we actually corrected in Figures 2-3. We thus prefer to keep the current color coding not to create other incompatibilities throughout the manuscript.
10. Line 263, it should be “ibuprofen”
Ans: Corrected throughout the manuscript.
11. Table 3, this is just a technical issue, you should try to fit “Temperature, K” in one line
Ans: Corrected.
12. Line 218, it should be “ibuprofen”
Ans: Corrected throughout the manuscript.
13. Figure 1, why only one enantiomer of ibuprofen is present here?
Ans: In our model, both enantiomers are described by identical force-field parameters. Configuration at the optical center is defined with the molecular constitution, which cannot change during the simulation once the simulation box is created. Note that the potentially conflicting dihedral parameters around the chiral center were set to zero (e. g. CA-CA-CT-HC, or OC-CO-CT-HC), or to a three-fold cosine function treating both ± 60° torsions equivalently (e. g. HC-CT-CT-HC). As the same atomic sites responsible for the hydrogen bonding are found in both enantiomers, we prefer to keep a single image of ibuprofen in Figure 1 for the sake of brevity.
14. General question. I understand that you have wanted to test the new method of calculations in order to predict the Tg of binary systems (this was one of the aim of this work). So, why have you chosen the models for that the experimental data is limited? I.e., Figure 2. It would be more meaningful if you could compare those computational results with the corresponding experimental ones (measuring the Tg of the mixtures of those API with either Ade or Cys in different molar ratio). If you can’t compare this with the experimental results you don’t really know how far from the “true” values are you.
Ans: We agree that a larger set of experimental data would enable to improve the statistical significance of our comparison of the calculated and experimental Tg values. To assess the accuracy of the underlying MD simulations for Tg predictions, we selected the particular API molecules due to the availability of experimental Tg and other properties. Thanks to this analysis of the accuracy of our predictions for pure compounds, we have a clear image about the accuracy of MD for such materials and models used. We also have a reasonable image what the accuracy for the mixtures could be. Provided that our error of Tg ranges around 60 K, we assume that the computational accuracy is similar for pure compounds and mixtures in this particular case. Unfortunately, any experimental data are extremely scarce for multi-component systems, including mixtures of API with biomolecules. Furthermore, our intention was to perform a computational study at first to identify promising candidate systems, thus to narrow the scope of subsequent experimental investigations of such amorphous dispersions, which we plan in next stages of our research of this topic. Fortunately, we already have a few unpublished experimental Tg data points which are relevant for this case. We added a comparison of our predictions with these new data and commented it in the manuscript accordingly.
15. Line 206, it should be „sophisticated”
Ans: Corrected.
16. Line 208, the other possible explanation is that, to properly model the Tg, you need to perform the ab initio MD, which, for such large systems is currently computationally non affordable.
Ans: Yes, we agree with the referee that ab initio MD for bulk liquids remain unaffordable. We mentioned this possible outlook in our conclusions.
17. Lines 347-349, this should be deleted.
Ans: Corrected.
18. Figure 8, aren’t there any experimental data in the literature to compare those results?
Ans: We thank for this suggestion for an improvement of our manuscript. We were able to find experimental data on vaporization enthalpies for naproxen and racemic ibuprofen. Our predictions differ by 1.5 and 5.5 kJ/mol, respectively, which indicates a reasonable accuracy of the underlying force-field models in terms of energetics of molecular interactions. We added a brief discussion of this aspect in our manuscript.
19. Lines 364-365, or the application of some QM methods, at least semi-empirical.
Ans: We thank for this suggestion. We incorporated it in our conclusions.
Reviewer 3 Report
I find the manuscript interesting for the readers, even though the authors could not achieve all the goals as intended. I think that a better overview of other published work in the field would help to improve the manuscript and put it into a general context.
As examples of important publications that deserve being discussed or at least mentioned I can list:
- Floudas, G., Paluch, M., Grzybowski, A., & Ngai, K. (2010). Molecular dynamics of glass-forming systems: effects of pressure (Vol. 1). Springer Science & Business Media.
- Bøhling, L., Ingebrigtsen, T. S., Grzybowski, A., Paluch, M., Dyre, J. C., & Schrøder, T. B. (2012). Scaling of viscous dynamics in simple liquids: theory, simulation and experiment. New Journal of Physics, 14(11), 113035.
- Rams-Baron, M., Jachowicz, R., Boldyreva, E., Zhou, D., Jamroz, W., & Paluch, M. (2018). Amorphous Drugs, Springer, Cham.
- Knapik-Kowalczuk, J., Tu, W., Chmiel, K., Rams-Baron, M., & Paluch, M. (2018). Co-stabilization of amorphous pharmaceuticals—The case of nifedipine and nimodipine. Molecular pharmaceutics, 15(6), 2455-2465.
- Koperwas, K., Adrjanowicz, K., Wojnarowska, Z., Jedrzejowska, A., Knapik, J., & Paluch, M. (2016). Glass-forming tendency of molecular liquids and the strength of the intermolecular attractions. Scientific reports, 6(1), 1-10.
- Phan, A. D., Knapik-Kowalczuk, J., Paluch, M., Hoang, T. X., & Wakabayashi, K. (2019). Theoretical model for the structural relaxation time in coamorphous drugs. Molecular pharmaceutics, 16(7), 2992-2998.
- Grzybowska, K., Grzybowski, A., Knapik-Kowalczuk, J., Chmiel, K., Woyna-Orlewicz, K., Szafraniec-Szczęsny, J., ... & Paluch, M. (2020). Molecular Dynamics and Physical Stability of Ibuprofen in Binary Mixtures with an Acetylated Derivative of Maltose. Molecular Pharmaceutics, 17(8), 3087-3105.
Author Response
Rev #3:
I find the manuscript interesting for the readers, even though the authors could not achieve all the goals as intended. I think that a better overview of other published work in the field would help to improve the manuscript and put it into a general context.
As examples of important publications that deserve being discussed or at least mentioned I can list:
- Floudas, G., Paluch, M., Grzybowski, A., & Ngai, K. (2010). Molecular dynamics of glass-forming systems: effects of pressure (Vol. 1). Springer Science & Business Media.
- Bøhling, L., Ingebrigtsen, T. S., Grzybowski, A., Paluch, M., Dyre, J. C., & Schrøder, T. B. (2012). Scaling of viscous dynamics in simple liquids: theory, simulation and experiment. New Journal of Physics, 14(11), 113035.
- Rams-Baron, M., Jachowicz, R., Boldyreva, E., Zhou, D., Jamroz, W., & Paluch, M. (2018). Amorphous Drugs, Springer, Cham.
- Knapik-Kowalczuk, J., Tu, W., Chmiel, K., Rams-Baron, M., & Paluch, M. (2018). Co-stabilization of amorphous pharmaceuticals—The case of nifedipine and nimodipine. Molecular pharmaceutics, 15(6), 2455-2465.
- Koperwas, K., Adrjanowicz, K., Wojnarowska, Z., Jedrzejowska, A., Knapik, J., & Paluch, M. (2016). Glass-forming tendency of molecular liquids and the strength of the intermolecular attractions. Scientific reports, 6(1), 1-10.
- Phan, A. D., Knapik-Kowalczuk, J., Paluch, M., Hoang, T. X., & Wakabayashi, K. (2019). Theoretical model for the structural relaxation time in coamorphous drugs. Molecular pharmaceutics, 16(7), 2992-2998.
- Grzybowska, K., Grzybowski, A., Knapik-Kowalczuk, J., Chmiel, K., Woyna-Orlewicz, K., Szafraniec-Szczęsny, J., ... & Paluch, M. (2020). Molecular Dynamics and Physical Stability of Ibuprofen in Binary Mixtures with an Acetylated Derivative of Maltose. Molecular Pharmaceutics, 17(8), 3087-3105.
Ans: We thank for a positive evaluation of our work, and also for the kind suggestions for additional literature. We selected the most recent and relevant works (5 out of 7 suggested) and incorporated mentions about the context of those in our introduction.
Round 2
Reviewer 2 Report
The Authors have corrected their manuscript, following my suggestions. Current version can be accepted for publication.